# Four-Year Environmental Surveillance Program of *Legionella* spp. in One of Palermo’s Largest Hospitals

**DOI:** 10.3390/microorganisms10040764

**Published:** 2022-04-01

**Authors:** Ignazio Arrigo, Elena Galia, Teresa Fasciana, Orazia Diquattro, Maria Rita Tricoli, Nicola Serra, Mario Palermo, Anna Giammanco

**Affiliations:** 1Unit of Microbiology, Virology and Parasitology, A.O.U. Paolo Giaccone, Via del Vespro 133, 90127 Palermo, Italy; mariaritatricoli@gmail.com; 2Legionella Reference Laboratory, University of Palermo, 90127 Palermo, Italy; teresa.fasciana@virgilio.it (T.F.); anna.giammanco@unipa.it (A.G.); 3Department of Health Promotion, Mother and Child Care, Internal Medicine and Medical Specialities, University of Palermo, 90127 Palermo, Italy; 4Laboratory of Microbiology, A. O. Ospedali Riuniti “Villa Sofia-Cervello”, 90100 Palermo, Italy; orazia.diquattro@villasofia.it; 5Department of Public Health, University Federico II of Naples, 80131 Napoli, Italy; nicola.serra5@gmail.com; 6Sicilian Health Department, Public Health and Environmental Risks Service, 90127 Palermo, Italy; mario.palermo1955@regione.sicilia.it

**Keywords:** *Legionella*, surveillance, water system, hospital

## Abstract

*Legionella* is a ubiquitous bacterium that lives in freshwater environments and colonizes human-made water systems. *Legionella pneumophila* is the most virulent species, and risk factors for Legionnaires’ disease include increasing age, smoking, chronic diseases, and immunodeficiency. For this reason, it is very important to assess and monitor hospital water systems in order to prevent legionellosis. We have monitored a large hospital in Palermo for four years. To determine the presence of microorganisms, according to national guidelines, we used the culture method, which is considered the gold standard for *Legionella* detection. Sampling was divided into five macro-areas, and a total of 251 samples were collected during the period of investigation, 49% of which were *Legionella* spp.-positive and 51% were *Legionella* spp.-negative. Positive samples with *L. pneumophila*. sgr 2-15 were most frequent in the Underground (55.6%, *p* = 0.0184), Medicine (42.9%, *p* = 0.0184) and Other (63.2%, *p* = 0.002) areas; while positive samples for *L. pneumophila* sgr 1 were less frequent in the Underground (0.0%, *p* = 0.0184) and Surgery areas (4.5%, *p* = 0.033), and for *Legionella anisa*, were less frequent in the Medicine (4.1%, *p* = 0.021), Oncohematology (0.0%, *p* = 0.0282), and Other (0.0%, *p* = 0.016) areas. Finally, no significant differences were observed among the areas for each isolate considered. The surveillance carried out in these years demonstrates the importance of monitoring, which allows us to analyze the conditions of hospital facilities and, therefore, prevent *Legionella* spp. infections.

## 1. Introduction

The genus *Legionella* includes gram-negative bacteria and was first detected in 1976 during a convention of the American Legion in a hotel in Philadelphia [1]. The genus includes over 60 species and more than 70 serogroups [2]. *L. pneumophila* is considered the most pathogenic species of the 16 serogroups, of which serogroup 1 is responsible for most cases of Legionnaires’ disease or Legionellosis, followed by *L. anisa*. Other *Legionella* species are rarely pathogenic in humans, the most common being *Legionella micdadei*, *Legionella bozemanii*, and *Legionella longbeachae* [3]. In nature, its presence is detected in ponds, lakes, rivers, lake surfaces, and wetlands in general. The microorganism passes from its natural reservoirs into the water distribution network, where the main source of contamination can be found [4]. *Legionella* spp. proliferates mainly in warm water between 25 °C and 45 °C with an optimal growth temperature between 37 °C and 42 °C, colonizing cooling towers, showers, hot springs, soils, and forming biofilms [5]. Deteriorated old pipes, water stagnation, and corroded pipes can generate favorable conditions for the proliferation of the microorganisms. In particular, a major problem concerns the formation of biofilm that can increase the ability to resist disinfection treatments. Legionnaires’ disease is an atypical pneumonia that occurs after an incubation period of 2–10 days. In addition, extrapulmonary symptoms such as headaches, muscle aches, and gastrointestinal disorders may occur. The bacterium could also cause Pontiac fever with flu-like symptoms, a less severe form of the illness associated with fever [6]. In Italy, since 1990, legionellosis has been included in the list of infectious diseases subject to mandatory notice [7]. The case fatality rate in Europe in 2018 was 8%, but higher percentages, between 15% and 34%, have been reported among the most vulnerable patients and hospitalized cases [8].

Our laboratory, recognized as a Regional Reference Center for Environmental and Clinical Surveillance and Control for Legionellosis in Western Sicily, according to ministerial guidelines, carries out surveillance in hospital buildings, with a special focus on the wards with immunocompromised patients.

We report the data achieved from the analysis of a four-year surveillance in a hospital in Palermo. The sampling was performed in various wards grouped into five areas, and specifically, the water sampling sites were boilers, storage tanks, sinks, and showers. The analysis showed a similar rate of positivity in all wards except in the Underground area, characterized by a lower number of positive samples. Our data point out a predominance of *L. pneumophila* sgr 2-15 in all areas, where, in addition, there was also the coexistence of *L. pneumophila* sgr 1, *L. pneumophila* sgr 2-15, and *Legionella* spp.

The results we have achieved reveal a rather steady bacterial load even during a period of 4 years despite the disinfection treatments used. This is because it is very difficult to permanently eradicate *L. pneumophila*, especially when it forms biofilms. The future goal is to continue with surveillance by examining the local microbiological distribution.

## 2. Materials and Methods

### 2.1. Study Proposal

As mentioned by the European Centre for Disease Prevention and Control (ECDC), in 2019, Italy, together with France, Germany, and Spain, accounted for 71% of all notified cases, although their combined populations represented only about 50% of the EU/EEA population [9]. It is important to carry out a detailed environmental inspection because of the nosocomial outbreaks that are subject to mandatory notice. Therefore, in this study, we have performed a four-year surveillance program in a hospital, from 2018 to 2021. According to the 2015 Italian guidelines for the prevention and control of legionellosis, testing should be carried out on a yearly basis. We continue to monitor the various departments in the hospital. Approval by the Ethics Committee was obtained by Azienda Ospedaliera Universitaria Policlinico “P. Giaccone” of Palermo (protocol No. 07/2019).

### 2.2. Sampling Procedure

The surveillance program was performed in a large public hospital in Palermo (Sicily, Italy) built in the early 1900s. The facility over time has been extended and modernized. The sampling was carried out in almost all the buildings, but to better expose the findings, the wards were grouped into five areas. The water comes from the municipal pipeline, which is already chlorinated. If necessary, chlorine can be added to the storage tanks to ensure that the optimal range is maintained. The principal sites of sampling were municipal waters, storage tanks (cold water), and boilers (hot water). With regard to each ward, the most sampled sites (sinks and showers) were located in the nearest and most distal points of the floor to the main pipeline coming from the tanks. To monitor the conditions of the water network of the building, it was necessary to collect the water through two methods—pre-flush (the sample was drawn as soon as the tap is opened) and post-flush (the water was left to run for a few minutes before withdrawing the sample). The first method allowed for evaluating the water in normal conditions of use and estimating the risk for the patient; the second method allowed for evaluating the conditions of the water system and efficacy of the disinfection system [10]. Samples of hot and cold water were collected using 1-L sterile plastic bottles with sodium thiosulphate to neutralize free chlorine. The water temperature was measured for each sample with a digital infrared laser thermometer (Blackout Tech, New Delhi, India). Furthermore, the presence of free chlorine was measured through a chlorine meter (Aqualytic, Perlabo S.A.S., Catania, Italy).

The samples were transported to the laboratory at ambient temperature and analyzed on the day of collection.

### 2.3. Sample Analysis

As mentioned above, the samples were processed according to the standards of the Italian ministerial guidelines and ISO 11731:2017. Briefly, the water samples were concentrated to 10 mL of the original filtrates by filtration through a 0.2 µm cellulose membrane (Sartorius AG, Gottingen, Germany) using a vacuum pump; 5 mL of the concentrated sample was placed in a 50 °C water bath for 30 min to reduce contamination. Subsequently, 100 µL of the treated sample was cultured on BCYE and the same amount of the untreated sample was inoculated on another BCYE plate (Becton, Dickinson, Franklin Lakes, NJ, USA) and incubated for 10 days at 37 °C in a moist chamber with 5% CO_2_. To have an adequate representation of the colonies present in the sample, after 3 days of incubation, five subcultures of typical colonies were performed for each plate, where possible, on a selective medium (BCYE) and cysteine-free medium. The suspected colonies were confirmed using a latex agglutination test with polyvalent antisera (Oxoid Spa, Milan, Italy) that allowed for the identification of *L. pneumophila* and the *Legionella* spp. [11]. The other species belonging to the genus *Legionella* spp. were identified by matrix-assisted laser desorption/ionization-time of flight mass spectrometry (MALDI-TOF) [12]. The total number of *Legionellae* spp. (CFU) in the reference volume of the sample was estimated based on the number of typical colonies counted on the plate considered, taking into account both untreated and heat-treated plates. According to this method, the lower limit of detection was 100 CFU/L.

### 2.4. Statistical Analysis

The data are presented as numbers or percentages for the categorical variables. A binomial test was performed to compare two mutually exclusive proportions or percentages in groups.

The multiple comparison chi-square tests were used to define significant differences among the percentages. In this case, if the chi-square test was significant (*p*-value < 0.05), residual analysis with the Z-test was performed.

All tests with a *p*-value (*p*) < 0.05 were considered significant. The statistical analysis was performed using MATLAB Statistical Toolbox version 2008 (MathWorks, Natick, MA, USA) for Windows at 32 bit.

## 3. Results

Five areas were monitored over four years of time—Underground facility (storage tanks, boilers); Medicine (Internal Medicine, Urology, Neurology, Emergency Room, Intensive Care Unit, Cardiology, Pneumology, Infectious Diseases); Surgery (Dentistry, Plastic Surgery, Obstetrics–Gynecology); Oncohematology; Others (Nuclear Medicine, Pediatric Orthopedics, Hospice, Gastroenterology, Polyclinics, Pediatrics, Laboratories, Psychiatrics, Psychiatry, Clinics, Angiography, Transfusion Medicine).

A total of 251 water samples were collected; 123 (49%) were positive for *Legionella* spp. and 128 negative (51%). In particular, Table 1 shows that the Underground area had the lowest number of positive samples as compared to others (24.7%, *p* = 0.0003). Only in the Underground area, the number of positive samples was significantly lower than the negative samples (24.7% vs. 75.3%, *p* < 0.0001). Regarding the positive water samples, most of them were contaminated by *L. pneumophila* sgr 2-15, followed by *L. pneumophila* sgr 1. As reported in Table 1, the percentages of positive samples for *Legionella* spp. collected during the period of study were very similar in all areas, with the exception of the Underground area. Moreover, as shown in Table 2, positive samples with *L. pneumophila* sgr 2-15 were most frequent in the Underground (55.6%, *p* = 0.0184), Medicine (42.9%, *p* = 0.0184), and Other (63.2%, *p* = 0.002) areas, while positive samples for *L. pneumophila* sgr 1 were less frequent in the Underground (0.0%, *p* = 0.0184) and Surgery areas (4.5%, *p* = 0.033), and positive samples for *L. anisa* were less frequent in the Medicine (4.1%, *p* = 0.021), Oncohematology (0.0%, *p* = 0.0282), and Other (0.0%, *p* = 0.016) areas. Finally, no significant differences were observed among the areas for each isolate considered.

We analyzed the collected water samples only using the culture method, currently considered the gold standard.

As shown in Figure 1, *L. pneumophila* sgr 1 was detected in 27 samples with a load between 101–1000 CFU/L. In addition, for *L. pneumophila* sgr 2-15, most positive isolates were in the range 101–1000 CFU/L except one sample with a load >100,000 CFU/L. Finally, regarding different *Legionella* spp., we identified *L. anisa* present in 12 samples, most of them (10 samples) with a load corresponding to the lowest range of positivity.

Figure 2 shows the annual trend of the different *L. pneumophila* serogroups and *L. anisa*.

The trend of isolates detected during the period of surveillance shows that *L. pneumophila* sgr 2-15 was present every year. The emergence of *L. anisa* for the past two years was also highlighted.

From our data, 34.14% of the positive samples show the coexistence of different isolates (Table 2). This is probably due to the ubiquitous contamination of *Legionella* in the water systems.

As underlined in Figure 3, during 2018, there was a clear dominance of *L. pneumophila* sgr 1 and *L. pneumophila* sgr 2-15; no coexistence was detected in 2019. In 2020, in four samples, we found the simultaneous presence of *L. pneumophila* sgr 1, *L. pneumophila* sgr 2-15, and *L. anisa*. Finally, in 2021, the coexistence of *L. pneumophila* sgr 1 and *L. pneumophila* sgr 2-15 became prevalent again. The total number of samples does not correspond to that reported in Table 1 because we found different *Legionella* serogroups and species in the same sample.

## 4. Discussion

*Legionella* spp. is a ubiquitous microorganism that proliferates in natural and artificial water environments. It can colonize water distribution systems of buildings, where it is known to form biofilms on the surfaces, resulting in increased resistance to disinfection treatments [13]. Several factors influence *Legionella* spp. colonization and growth, including temperature, stagnation, age, and the material used in the pipes, which can facilitate the formation of corrosion that generates a favorable habitat for the colonization of the bacterium. In particular, healthcare facilities represent ideal reservoirs for *Legionella* spp. multiplication considering the complexity of the water distribution system [14]. Legionnaires’ disease is an atypical pneumonia caused by the inhalation of aerosols containing *Legionella* spp. [15]. It could be a health risk for hospitalized patients, especially in immunocompromised patients. For this reason, it is appropriate to carry out surveillance programs in nosocomial environments, mainly in the most critical departments. Indeed, the bacterium can spread not only through a water matrix but especially due to aerosols from air conditioning systems, cooling towers, and so forth [16]. Analyzing the data shown in Table 1, it should be noted that in the Underground area, there was a lower contamination than in the other areas. This is likely due to increased monitoring of the water in tanks and boilers that were more easily accessible. It is not surprising that levels of significant contamination have been found in the other areas where there were old water pipelines in which most likely fouling and dead points with water stagnation can be found. On the other hand, the large size of the hospital water network makes it impossible to maintain an adequate level of chlorination to inhibit bacterial proliferation [17]. Regarding other *Legionella* spp., such as *L. anisa*, previous studies have shown that it may be more difficult to detect them in water than *L. pneumophila*, and their true prevalence may be somewhat underestimated. There is evidence that some *Legionella* spp. proliferate less readily in the presence of the sediment and commensal microflora of water systems than *L. pneumophila* [18]. As pointed out in other studies [19], the isolation of other *Legionella* species can be difficult, especially if there is a high load of *L. pneumophila*, as the latter tends to predominate. To the best of our knowledge, there is little information on the diversity of hospital water supply contaminated by *Legionella* spp. and other species. Despite the surveillance programs periodically carried out, a significant decrease in the bacterial load has not been recorded. The difficulty in eradicating the microorganism most probably is due to the old water network of the hospital, where the bacterium finds ideal conditions for survival. Therefore, during these years of surveillance, it has been possible to detect a high load in the various areas; *L. pneumophila* sgr 2-15 has particularly shown a constant trend over the years. The disinfection system adopted by the hospital involves periodic chlorination. However, this system has a limited effect in time, and the eventual formation of biofilm involves an increasing resistance to this chemical treatment. Since the definitive elimination of the microorganism is extremely difficult, a suitable system to avoid the spread and contamination, especially in the most critical wards, could include the use of anti-Legionella filters that are easy to mount and to keep under control. This recommendation helps to decrease the incidence of all nosocomial waterborne infections [20].

## 5. Conclusions

The four-year surveillance program carried out highlights the importance of monitoring that allows for the analysis of the conditions of the hospital’s water system. The analysis shows that the load of microorganisms tends to be rather constant over time despite the disinfection processes and controls carried out by our reference center. *Legionella* spp. colonization of hospital water systems is no longer new, and its control is very difficult as disinfection methods are effective in the short-term but not in the long term, especially in hospitals with superannuated water systems [21]. The complete eradication of the germ is complicated for its ability to resist the most common physical and chemical disinfection treatments, especially where it forms biofilms. Our data, based on the annual trend and bacterial load, show that the *L. pneumophila* sgr 2-15 strains are particularly resistant to water disinfection processes and, therefore, difficult to eradicate from the hospital water supply. In addition, the emergence of *L. anisa* increases the possibility of infection in hospitalized patients. Thus, it is very important to continue with the surveillance program, especially if we consider that in the hospital buildings, there are many immunocompromised patients with a very high risk of getting legionellosis. Moreover, although hospital surveillance is carried out periodically, as indicated in the Italian ministerial guidelines, the general situation on the island is not adequately monitored. In Sicily, the rate of notifications is not high, especially when compared to Northern Italy [22]. It is precisely for these reasons that it is essential to continue with the investigations so that we can correctly assess the epidemiological condition of our territory. To reduce the chance of Legionnaires’ disease transmission in healthcare facilities, the CDC (Centre for Disease Control and Prevention) recommends a strategy that focuses on the proper maintenance of water systems by the application of anti-Legionella filters [23,24]. It would also be helpful to overhaul the entire water system and replace the damaged pipelines that cause the proliferation of the bacterium.

## 6. Study Limitations

This study represents the first analysis carried out in a large hospital in our city. The absence of a long-term follow-up prevents the availability of a large amount of data. The limitation related to the lack of surveillance in the metropolitan area encourages us to continue and extend the investigations.

## Figures and Tables

**Figure 1 microorganisms-10-00764-f001:**
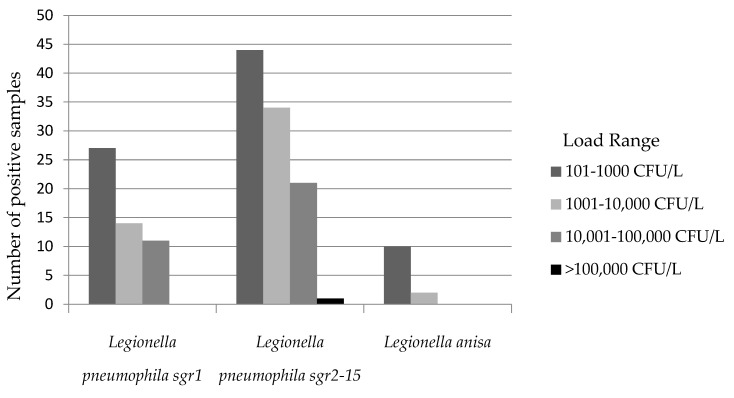
Total number of positive samples related to the bacterial load of *L. pneumophila*. and *L. anisa* isolates. For each of the three clusterings (*L. pneumophila* sgr 1, *L. pneumophila* sgr 2-15, *L. anisa*), the greatest number of positive samples had a load within the lowest range, the value of which has been established by Italian ministerial guidelines.

**Figure 2 microorganisms-10-00764-f002:**
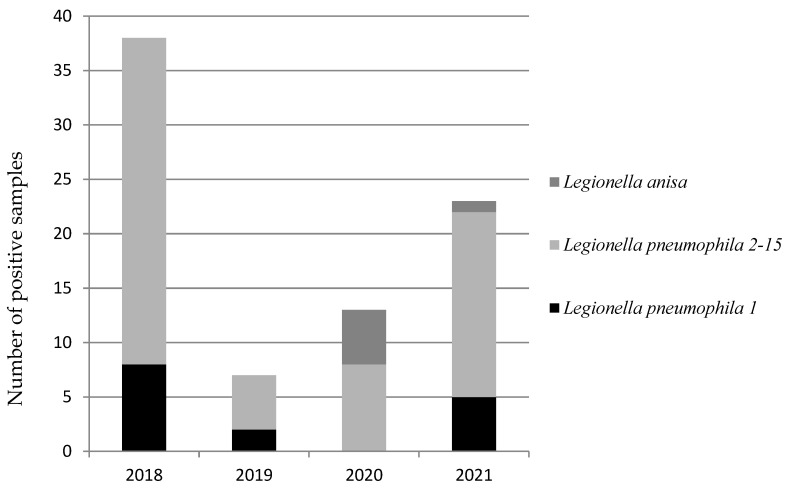
Trend recorded on a yearly basis during the four-year surveillance program reporting the number of positive isolates for *L. pneumophila* serogroups and *L. anisa*.

**Figure 3 microorganisms-10-00764-f003:**
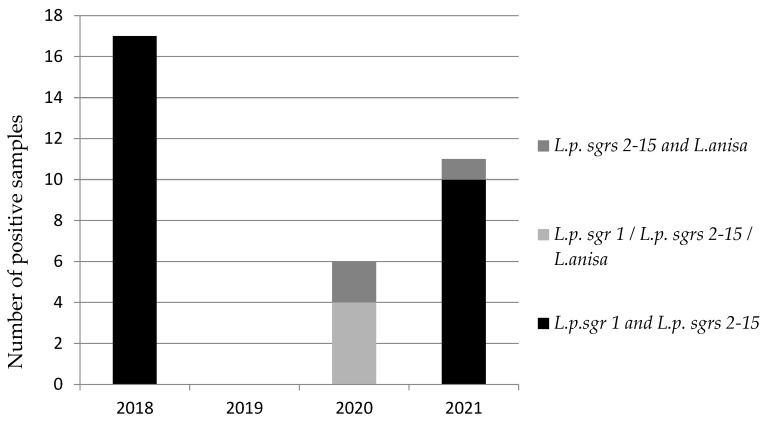
Number of samples with presence of co-isolates during the four-year surveillance program, regarding different *L. pneumophila (L.p.)* sgrs and *L. anisa*.

**Table 1 microorganisms-10-00764-t001:** The rates of positive and negative samples for *Legionella* spp.

Area	Total	Underground (Und)	Medicine (Med)	Surgery (Sur)	Oncohematology (Onc)	Others (Oth)	Analysis among Areas *p*-Value (Test)
Positive samples	123 (49%)	18 (24.7%)	49 (60.5%)	22 (53.7%)	15 (62.5%)	19 (59.4%)	*p* < 0.0001 * (C) Und (+) ***, *p* = 0.0003 (Z)
Negative samples	128 (51%)	55 (75.3%)	32 (39.5%)	19 (46.3%)	9 (37.5%)	13 (40.6%)	
Total samples	251	73	81	41	24	32	
Positive vs. Negative	*p* = 0.75 (B)	*p* < 0.0001 * (B)	*p* = 0.06 (B)	*p* = 0.64 (B)	*p* = 0.22 (B)	*p* = 0.29 (B)	

* = statistically significant test; + = positive samples; *** = less frequent; C = multiple chi-square test; Z = post hoc Z-test; B = binomial test.

**Table 2 microorganisms-10-00764-t002:** Distribution of *L. pneumophila* sgr 1 and sgr 2-15, of *L. anisa* and isolates in the five areas.

Area	Positive for *L. pneumophila* sgr 1 sgr 2-15		Positive for *L. anisa*	Co-Isolates	Analysis for Each Area *p*-Value (Test)
Underground (Und)	0.0% (0/18)	55.6% (10/18)	11.1% (2/18)	33.3% (6/18)	*p* = 0.0006 * (C) sgr 2-15 **, *p* = 0.0184 (Z) sgr 1 ***, *p* = 0.0184 (Z)
Medicine (Med)	20.4% (10/49)	42.9% (21/49)	4.1% (2/49)	32.7% (16/49)	*p* < 0.0001 * (C) sgr 2-15 **, *p* = 0.0184 (Z) *L. anisa* ***, *p* = 0.021 (Z)
Surgery (Sur)	4.5% (1/22)	45.5% (10/22)	9.1% (2/22)	40.9% (9/22)	*p* = 0.0013 * (C) sgr 1 ***, *p* = 0.033 (Z)
Oncohematology (Onc)	6.7% (1/15)	46.7% (7/15)	0.0% (0/15)	46.7% (7/15)	*p* = 0.0017 * (C) *L. anisa* ***, *p* = 0.0282 (Z)
Others (Oth)	15.8% (3/19)	63.2% (12/19)	0.0% (0/19)	21.1% (4/19)	*p* < 0.0001 * (C) sgr 2-15 **, *p* = 0.002 (Z) *L. anisa* ***, *p* = 0.016 (Z)
Cross-area analysis for positive samples	*p* = 0.11 (C)	*p* = 0.60 (C)	*p* = 0.38 (C)	*p* = 0.56 (C)	
Total (positive samples)	12.2% (15/123)	48.8% (60/123)	4.9% (6/123)	34.1% (42/123)	*p* < 0.0001 * (C) sgr 2-15 **, *p* < 0.0001 (Z) sgr 1 ***, *p* = 0.0034 (Z) *L. anisa* ***, *p* < 0.0001 (Z)

* = statistically significant test; ** = more frequent; *** = less frequent; C = multiple chi-square test; Z = post hoc Z-test.

## Data Availability

Not applicable.

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
