# Peer review of "Four-Year Environmental Surveillance Program of Legionella spp. in One of Palermo’s Largest Hospitals"

_microorganisms, 2022, doi:10.3390/microorganisms10040764_

Round 1

Author Response

Dear Reviewer,

Reviewer 2 Report

I have previously suggested and I keep my suggestion regarding the antimicrobial profiles of isolated Legionella strains. 

Author Response

Dear Reviewer,

Reviewer 3 Report

The corrected version of the article is much better than the original one. The authors have improved and strengthened many aspects of this work. Currently, I have no comments and believe that the article can be published.

Author Response

Dear Reviewer,

Thank you very much

This manuscript is a resubmission of an earlier submission. The following is a list of the peer review reports and author responses from that submission.

Round 1

Reviewer 1 Report

The study was a good initiative to evaluate the water systems of a hospital building for Legionella.

Comments:

Line 18: capable of causing, 18 mainly in immunocompromised people – may not be a true statement.

Line 20: “nosocomial water systems” nosocomial mainly used for disease originated from a hospital setting. Better to use “hospital water systems”.

Line 22: may be “culture method” instead of “culture survey”

Line 40: In nature we detect its presence in ponds, lakes, rivers, lake surfaces and wetlands in general. – passive sentence would be better and remove “we”.

Line 44: “Legionella multiplies 43 mainly in hot water from 30 to 40°C with an optimum at 32°C” – should check with other references. Most cited optimum growth temperature for Legionella is higher than 32 °C.

(van der Kooij, D., Brouwer-Hanzens, A. J., Veenendaal, H. R., & Wullings, B. A. (2016). Multiplication of Legionella pneumophila Sequence Types 1, 47, and 62 in Buffered Yeast Extract Broth and Biofilms Exposed to Flowing Tap Water at Temperatures of 38°C to 42°C. Applied and environmental microbiology82(22), 6691–6700. https://doi.org/10.1128/AEM.01107-16)

Line 45: Old pipes made of deteriorating materials? May be corroded or deteriorated old pipe

Line 59: nosocomial buildings/ hospital buildings

Line 74: ECDC needs the full name once at least

Line 149: Figure 1: Need to italicize the bacterial name

Results:

The author used culture method and spread 2 portions of 100 µl (with heat treatment and without heat treatment) on BCYE plate but never mentioned how they use the data (from which plate). Since the plate had no antibiotics, other bacteria should grow. Moreover, only five subcultures from plate were made. Therefore, it is difficult to understand how the count of the Legionella and different species/serogroups in cfu/L is calculated.

Discussion and conclusion:

The authors need to discuss the results and limitation of the study and correlate the results with the current state of knowledge in the subject area. Should not make a general discussion and same is true for the conclusion which has to be very specific interpretation of the results, future direction and importance of the study.

Reviewer 2 Report

In the surveillance report of ECDC (2019), Legionnaires’ disease remains an uncommon and mainly sporadic respiratory infection, with annual notification rate increased in recent years, from 1.4 in 2015 to 2.2 cases per 100 000 populations in 2019. For this point of view, the subject of the article is epidemiologically topical, but the study has serious flaws and I strongly recommend to reject the article.

Reviewer 3 Report

This study is of interest to epidemiologists and organizers of the treatment process. The data obtained do not have much scientific novelty. Due to the large amount of collected material, this work will be interesting for a wide range of medical professionals. I believe that it can be published in the presented form.